# PhANNs, a fast and accurate tool and web server to classify phage structural proteins

Vito Adrian Cantu[1,2], Peter Salamon[2,3], Victor Seguritan[1¤a], Jackson Redfield[4¤b], David Salamon[3], Robert A. Edwards[1,2,4¤c], Anca M. Segall[1,2,4]*

1 Computational Science Research Center, San Diego State University, San Diego, United States of America, 2 Viral Information Institute, San Diego State University, San Diego, United States of America, 3 Department of Mathematics and Statistics, San Diego State University, San Diego, United States of America, 4 Department of Biology, San Diego State University, San Diego, United States of America

¤a Current address: Experian, Costa Mesa, CA, United States of America
¤b Current address: Inova Diagnostics, San Diego, CA, United States of America
¤c Current address: College of Science and Engineering, Flinders University, South Australia
* asegall@sdsu.edu

**Data Availability Statement:** The full database, as well as the code for PhANN and the webserver, are available for download at http://edwards.sdsu.edu/

## Abstract

For any given bacteriophage genome or phage-derived sequences in metagenomic data sets, we are unable to assign a function to 50–90% of genes, or more. Structural protein-encoding genes constitute a large fraction of the average phage genome and are among the most divergent and difficult-to-identify genes using homology-based methods. To understand the functions encoded by phages, their contributions to their environments, and to help gauge their utility as potential phage therapy agents, we have developed a new approach to classify phage ORFs into ten major classes of structural proteins or into an "other" category. The resulting tool is named PhANNs (Phage Artificial Neural Networks). We built a database of 538,213 manually curated phage protein sequences that we split into eleven subsets (10 for cross-validation, one for testing) using a novel clustering method that ensures there are no homologous proteins between sets yet maintains the maximum sequence diversity for training. An Artificial Neural Network ensemble trained on features extracted from those sets reached a test $F_1$-score of 0.875 and test accuracy of 86.2%. PhANNs can rapidly classify proteins into one of the ten structural classes or, if not predicted to fall in one of the ten classes, as "other," providing a new approach for functional annotation of phage proteins. PhANNs is open source and can be run from our web server or installed locally.

## Author summary

Bacteriophages (phages, viruses that infect bacteria) are the most abundant biological entity on Earth. They outnumber bacteria by a factor of ten. As phages are very different from each other and from bacteria, and we have relatively few phage genes in our database compared to bacterial genes, we are unable to assign function to 50–90% of phage genes. In this work, we developed PhANNs, a machine learning tool that can classify a phage

phanns and https://github.com/Adrian-Cantu/
PhANNs.

**Funding:** This research is based upon work
supported by the Office of the Director of National
Intelligence (ODNI), Intelligence Advanced
Research Projects Activity (IARPA), via the Army
Research Office (ARO) under cooperative
Agreement Number W911NF-17-2-0105, and
awarded as a partial subcontract to AMS. The
views and conclusions contained herein are those
of the authors and should not be interpreted as
necessarily representing the official policies or
endorsements, either expressed or implied, of the
ODNI, IARPA, ARO, or the U.S. Government. The
U.S. Government is authorized to reproduce and
distribute reprints for Governmental purposes
notwithstanding any copyright annotation thereon.
This work was supported by NIH grant
RC2DK116713 to AMS and RAE and US
Department of Defense: Defense Threat Reduction
Agency grant number DTRA13081-32220 to RAE.
Victor Seguritan and Jackson Redfield were
supported by NSF DMS 0827278 Undergraduate
BioMath Education grant awarded to AMS and PS.
The funders had no role in study design, data
collection and analysis, decision to publish, or
preparation of the manuscript.

**Competing interests:** The authors have declared
that no competing interests exist.

gene as one of ten structural roles, or "other". This approach does not require a similar
gene to be known.

This is a *PLOS Computational Biology* Software paper.

## Introduction

Bacteriophages (phages) are the most abundant biological entity on the Earth [1]. They modulate microbial communities in several possible ways: by lysing specific taxonomic members or narrow groups of microbiomes, they affect the microbial population dynamics and change niche availability for different community members. Via transduction and/or lysogeny, they mediate horizontal transfer of genetic material such as virulence factors [2], metabolic auxiliary genes [3], photosystems and other genes to enhance photosynthesis[4], and phage production in general, by providing the host with immunity from killing by other phages. Temperate phages can become part of the host genome as prophages; most bacterial genomes contain at least one, and often multiple prophages [5,6].

Phage structures (virions) are composed of proteins that encapsulate and protect their genomes. The structural proteins (or virion proteins) also recognize the host, bind to its surface receptors and deliver the phage's genome into the host's cell. Phage proteins, especially structural ones, vary widely between phages and phage groups, so much so that sequence alignment based methods to assign gene function fail frequently: we are currently unable to assign function to 50–90% of phage genes [7]. Experimental methods such as protein sequencing, mass spectrometry, electron microscopy, or crystallography, in conjunction with antibodies against individual proteins, can be used to identify structural proteins but are expensive and time-consuming. A fast and easy-to-use computational approach to predict and classify phage structural proteins would be highly advantageous as part of pipelines for identifying functional roles of proteins of bacteriophage origins. The current increased interest in using phages as therapeutic agents [8,9] motivates annotations for as much of the phage genome as possible. Even if they are somewhat tentative and not experimentally validated, annotations of the relatively non-toxic structural proteins versus the potentially host health-threatening toxins and other virulence factors could inform decisions whether to choose one specific phage versus another.

Machine learning has been used to attack similar biological problems. In 2012, Seguritan et al. [10] developed Artificial Neural Networks (ANNs) that used normalized amino acid frequencies and the theoretical isoelectric point to classify viral proteins as structural or not structural with 85.6% accuracy. These ANNs were trained with proteins of viruses from all three domains of life. They also trained two distinct ANNs to classify phage capsid versus phage non-capsid ORFs and phage "tail associated" versus phage "non-tail-associated" ORFs. Subsequently, several groups have used different machine learning approaches to improve the accuracy of predictions. The resulting tools are summarized in **Table 1**.

Each of these previous approaches has important limitations: 1) The classification is limited to two classes of proteins (e.g.,"capsid" or "not capsid"). 2) Training and testing sets were small (only a few hundred proteins in some cases), limiting the utility of these approaches beyond those proteins used in testing. 3) Methods that rely on predicting secondary structure (e.g., VIRALpro [11]) are slow to run. In general, these newer methods have improved accuracy at the cost of lengthening the time required for training, or have used very small training and/or test sets.

**Table 1. Summary of previous ML-based methods for classifying viral structural proteins.**

| Reference | Method | Target proteins | Database size | Accuracy |
|---|---|---|---|---|
| Seguritan et al.[10] | ANN | structural (all viruses) versus non-structural (all viruses) | 6,303 structural | 85.6% |
| | | | 7,500 non-structural | |
| Seguritan et al.[10] | ANN | capsid versus non-capsid (phages only) | 757 capsid | 91.3% |
| | | | 10,929 non-capsid | |
| Seguritan et al.[10] | ANN | Tail-associated versus non-tail (phages only) | 2,174 tail | 79.9% |
| | | | 16,881 non-tail | |
| Feng et al.[33] | Naïve Bayes | structural versus non-structural | 99 structural | 79.15% |
| | | | 208 non-structural | |
| Zhang et al.[34] | Ensemble Random Forest | structural versus non-structural | 253 structural | 85.0% |
| | | | 248 non-structural | |
| Galiez et al.[11] | SVM | capsid versus non-capsid | 3,888 capsid | 96.8% |
| | | | 4,071 non-capsid | |
| Galiez et al.[11] | SVM | tail versus non-tail | 2,574 tail | 89.4% |
| | | | 4,095 non-tail | |
| Manavalan et al.[35] | SVM | structural versus non-structural | 129 structural | 87.0% |
| | | | 272 non-structural | |
| This work | ANN | Ten distinct phage structural classes plus "others" | 168,660 structural | 86.2% |
| | | | 369,553 non-structural | |

Artificial Neural Networks (ANN) are proven universal approximators of functions in $\mathbb{R}^n$ [12], including the mathematical function that maps features extracted from a phage protein sequence to its structural class. We have constructed a manually-curated database of phage structural proteins and have used it to train a feed-forward ANN to assign any phage protein to one of eleven classes (ten structural classes plus a catch-all class labeled "others"). Furthermore, we developed a web server where protein sequences can be uploaded for classification. The full database, as well as the code for PhANNs and the webserver, are available for download at http://edwards.sdsu.edu/phanns and https://github.com/Adrian-Cantu/PhANNs

## Methods

### Database

In this work, we generated two complementary protein databases, "classes" and "others". The "classes" database contains curated sequences of ten phage structural functions (Major capsid, Minor capsid, Baseplate, Major tail, Minor tail, Portal, Tail fiber, Tail sheath, Collar, and Head-Tail Joining). These functional classes are not exhaustive (and we will add more classes in the future); they represent the dominant structural protein roles present in most (but not all) phages [13]. The terms/descriptors for these classes are addressed in the next section. Major capsid proteins are those that form the phage head. Many but not all phages also encode minor capsid proteins that decorate and/or stabilize the head or proteins present at the vertices of the icosahedral head or at the center of the hexon faces. Portals form a ring at the base of the phage head and serve to dock the packaging complex that translocates the genome into the phage head. Head-tail joining (aka head-tail connector or head completion) proteins form rings inserted between the portal ring and the tail. The collar is present in some phages, *e.g.* the Lactococcal phages, at the base of the neck/top of the tail to which the so-called whiskers attach. Major tail proteins form the inner tail tube of the tailed phages, whereas the tail sheath (aka the tail shaft) proteins form the outside of the tail, and permit contraction. Minor tail

proteins may comprise several kinds of proteins associated with the tail, including the tape measure protein. Baseplate proteins are those that are attached to the tail and to which the tail fibers are attached, the latter being a relatively common determinant of host range. The "others" database contains all phage ORFs that do not encode proteins annotated as "structural" or as any of the ten categories above.

### The database of "classes"

Sequences from the ten structural classes were downloaded from NCBI's protein database using a custom search for the class title (the queries are in the "ncbi_get_structural.py" script in the GitHub repository). Curation consisted of grouping sequences by their description (part of the fasta header) and deciding which descriptions to include. The list of included headers for each class can be found here https://github.com/Adrian-Cantu/PhANNs/tree/master/model_training/01_fasta; the variations of terms included are too many to be included here. All the terms preceded by a "+" (or "+ +") were included in the respective database. In the particular case of tail fibers, we did not include the descriptions "phage tail fiber assembly protein" (3,662 proteins) nor many "partial protein" variations (1,500+ proteins).

This method for collecting data has the limitation that a proportion of phage sequences in the database are misannotated and that NCBI has no controlled vocabulary for bacteriophage protein functions so it is occasionally difficult to account for misspelled annotations and/or alternative naming. However, it is clear from previous machine learning applications that a larger number of training examples is more important for optimal model performance than a perfectly curated training set [14]. To minimize inclusion of wrongly annotated protein sequences, we manually curated the databases to address these limitations.

### The "others" database

To generate a database for the "others" class, all available phage genomes (8,238) were downloaded from GenBank on 4/13/19. ORFs were found using the GenBank PATRIC [15] server with the phage recipe [16]. Sequences annotated as structural or any of the ten classes were removed during manual curation. Furthermore, the remaining sequences were de-replicated at 60% together with sequences in the "classes" database using CD-hit [17]. Any phage ORF that clustered with a sequence from the "classes" database was removed from the "others" database.

### Training, test, and validation split

Sequences in each class were clustered at 40% using CD-hit and split into eleven sets (10 for cross validation and one for testing, as shown in **Fig 1**). Once the clusters were established, to prevent loss of the sequence diversity available within the clusters, which is essential for optimal training, the clusters were expanded by adding back *within* each set all the representatives of that set (described in **Fig 1**). Subsequently, the sets corresponding to each structural class were merged. We named the generated sets 1D-10D and TEST. Splitting the database this way ensures that the different sets share no homologous proteins while recapturing all the sequence diversity present in each class. Finally, 100% dereplication was performed to remove identical sequences (See **Table 2**). The effect of the cluster expansion on performance is explored in **S1** and **S2** Figs.

### Extraction of features

The frequency of each dipeptide (400 features) and tripeptide (8,000 features) was computed for each ORF sequence in both the "classes" and "others" databases. As a potential time-saving

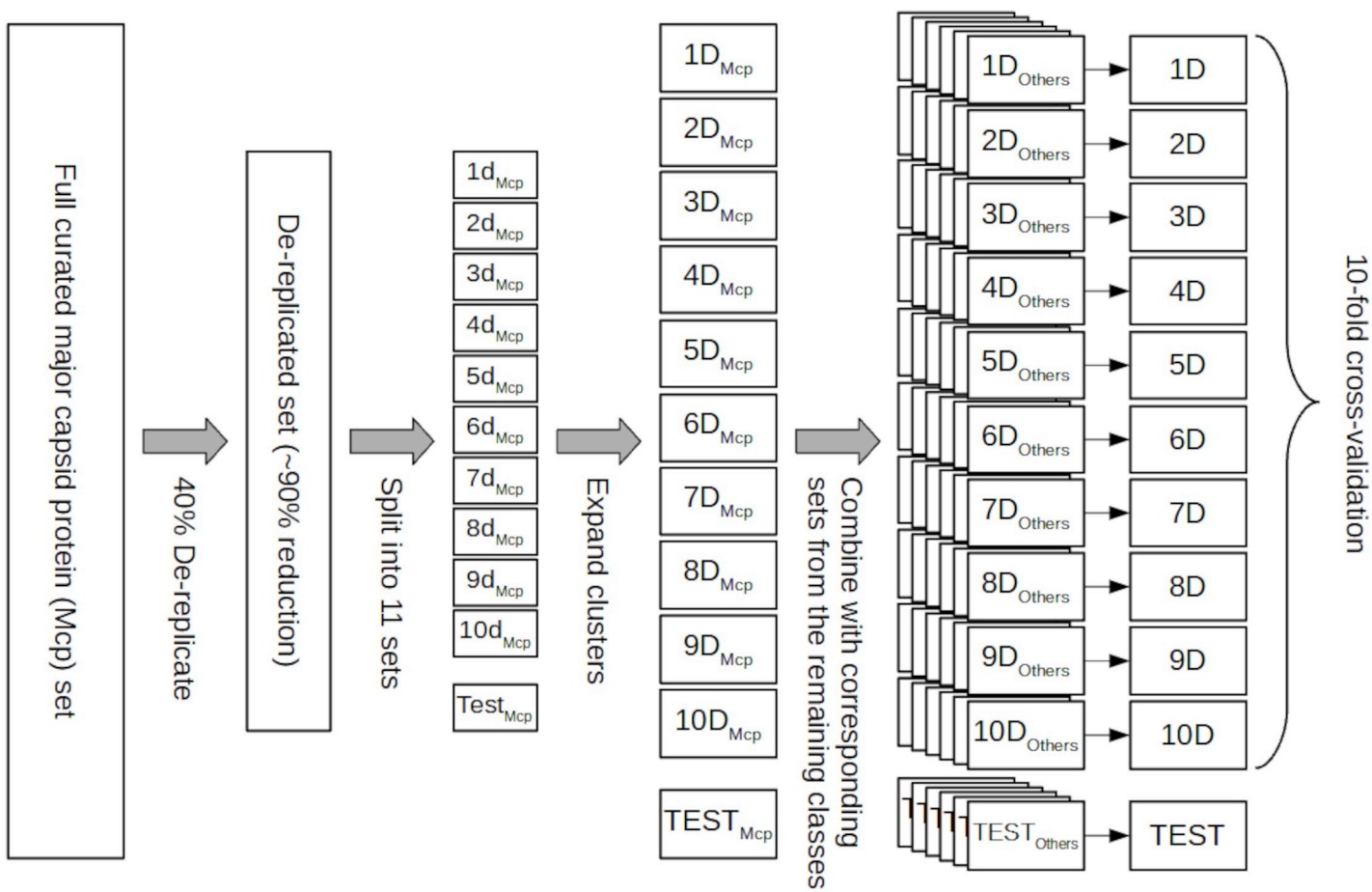

**Fig 1. Non homologous database split—To ensure that no homologous sequences are shared between the test, validation, and training sets the sequences from each class (Major capsid proteins in this figure) were de-replicated at 40%.** In the de-replicated set, no two proteins have more than 40% identity and each sequence is a representative of a larger cluster of related proteins. The de-replicated set is then randomly partitioned into eleven equal size subsets, ($1d_{Mcp}$-$10d_{Mpc}$ plus $Test_{Mpc}$). Those subsets are expanded by replacing each sequence with all the sequences in the cluster it represents (subsets $1D_{Mpc}$-$10D_{Mpc}$ plus $TEST_{Mpc}$). Analogous subsets are generated for the remaining ten classes and corresponding subsets are combined to generate the subsets used for 10-fold cross-validation and testing (1D-10D and TEST).

**Table 2. Database numbers—Raw sequences were downloaded using a custom script available at https://github.com/Adrian-Cantu/PhANNs.** All datasets can be downloaded from the web server. *Numbers before and after removing sequences at least 60% identical to a protein in the classes database.

| Class | Raw sequences | After manual curation | After de-replication at 40% | After expansion and de-replication at 100% |
|---|---|---|---|---|
| Major capsid | 112,987 | 105,653 | 1,945 | 35,755 |
| Minor capsid | 2,901 | 1,903 | 261 | 1,055 |
| Baseplate | 75,599 | 19,293 | 401 | 6,221 |
| Major tail | 66,513 | 35,030 | 536 | 7,704 |
| Minor tail | 94,628 | 80,467 | 918 | 18,002 |
| Portal | 210,064 | 189,143 | 2,310 | 59,745 |
| Tail fiber | 29,132 | 18,514 | 1,222 | 7,256 |
| Tail sheath | 37,885 | 35,570 | 599 | 15,349 |
| Collar | 4,224 | 3,709 | 339 | 2,105 |
| Head-Tail joining | 60,270 | 58,658 | 1,317 | 15,468 |
| **Total structural** | **694,203** | **547,940** | **9,848** | **168,660** |
| Others | 733,006 | 643,735/643,380* | 106,004 | 369,553 |

procedure during neural net training while also permitting classification of more diverse sequences, each amino acid was assigned to one of seven distinct "side chain" chemical groups (S1 Table). The frequency of the "side chain" 2-mers (49 features), 3-mers (343 features), and 4-mers (2,401 features) was also computed. Finally, some extra features, namely isoelectric point, instability index (whether a protein is likely to degrade rapidly; [18]), ORF length, aromaticity (relative frequency of aromatic amino acids; [19]), molar extinction coefficient (how much light the protein absorbs) using two methods (assuming reduced cysteins or disulfide bonds), hydrophobicity, GRAVY index (average hydropathy; [20]) and molecular weight, were computed using Biopython [21]. All 11,201 features were extracted from each of 538,213 protein sequences. The complete training data set can be downloaded from the web server (https://edwards.sdsu.edu/phanns).

### ANN architecture and training

We used Keras [22] with the TensorFlow [23] back-end to train eleven distinct ANN models using a different subset of features. We named the models to indicate which feature sets were used in training: the composition of 2-mers/dipeptides (di), 3-mers /tripeptides (tri) or 4-mer/ tetrapeptide (tetra), or side chain groups (sc) (as shown in S1 Table), and whether we included the extra features (p) or not. A twelfth ANN model was trained using all the features (Table 3).

Each ANN consists of an input layer, two hidden layers of 200 neurons, and an output layer with 11 neurons (one per class). A dropout function with 0.2 probability was inserted between layers to prevent overfitting. ReLU activation (to introduce non-linearity) was used for all layers except the output, where softmax was used. Loss was computed by categorical cross-entropy and the ANN is trained using the "opt" optimizer until 10 epochs see no training loss reduction. The model at the epoch with the lowest validation loss is used. Class weights inversely proportional to the number of sequences in that class were used.

**10-fold cross-validation.** Sets 1D to 10D (see Fig 1) were used to perform 10-fold cross-validation; ten ANNs were trained as described above, sequentially using one set as the validation set and the remaining nine as the training set. The results are summarized in Figs 2, 3, 4, S1 and S2.

**Table 3. Feature types included in each of the 12 models. di**—2-mer/dipeptide composition; **tri**—3-mer/tripeptide composition; **tetra**—4-mer/tetrapeptide composition; sc—side-chain grouping; **p**—plus all the extra features [isoelectric point, instability index (whether a protein is likely to be degraded rapidly), ORF length, aromaticity (relative frequency of aromatic amino acids), molar extinction coefficient (how much light a protein absorbs) using two methods (assuming reduced cysteines or disulfide bonds), hydrophobicity, GRAVY index (average hydropathy), and molecular weight, as computed using Biopython. - *Per class score figures are available as supplementary material.

| Model | di | tri | di_sc | tri_sc | tetra_sc | p |
|---|---|---|---|---|---|---|
| di_sc* | | | x | | | |
| di_sc_p* | | | x | | | x |
| tri_sc* | | | | x | | |
| tri_sc_p* | | | | x | | x |
| tetra_sc* | | | | | x | |
| tetra_sc_p* | | | | | x | x |
| di | x | | | | | |
| di_p | x | | | | | x |
| tri | | x | | | | |
| tri_p | | x | | | | x |
| tetra_sc_tri_p | | x | | | x | x |
| all | x | x | x | x | x | x |

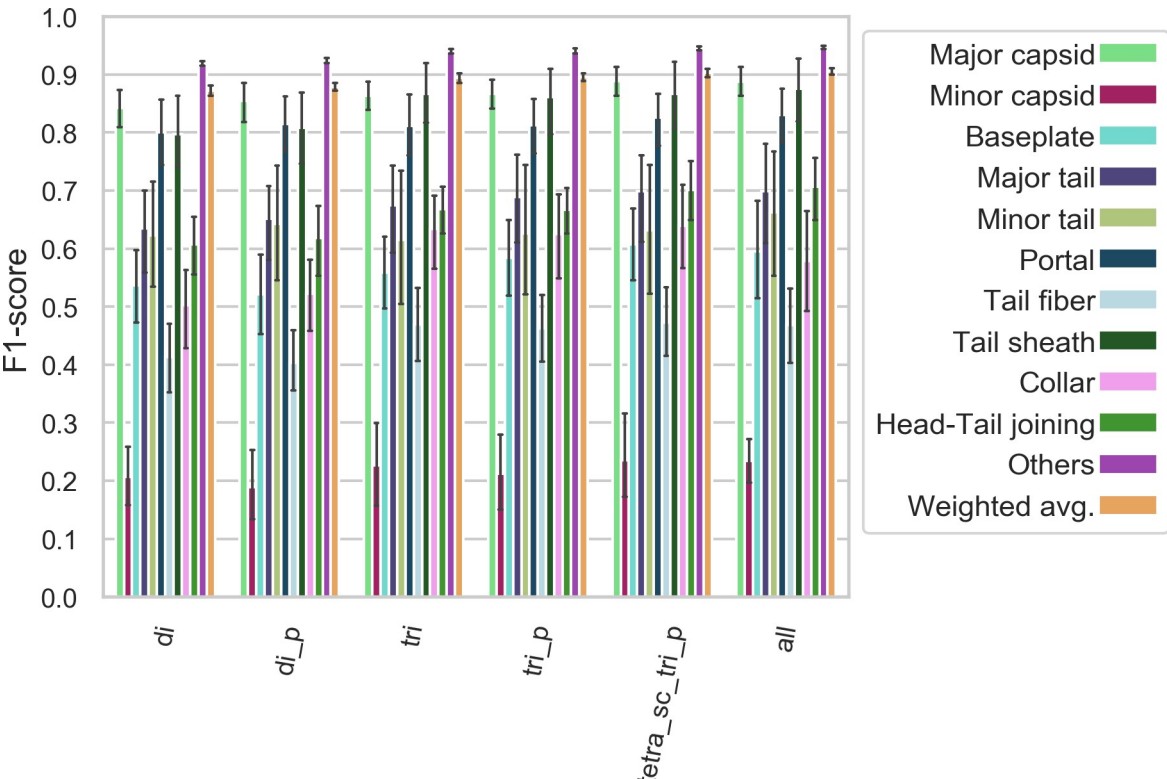

**Fig 2. Model-specific F$_1$ score—F$_1$ scores (harmonic mean of precision and recall) for each polypeptide model/class combination.** All models follow similar trends as to which classes are more or less difficult to classify correctly. Error bars represent the 95% confidence intervals.

**The PhANNs score.** For each input sequence, PhANNs run 10 ANNs predictions (those trained during the 10-fold cross-validation). Each of those 10 ANNs outputs the soft-max scores for every class (a number between 0 and 1, such that the score of all classes adds to 1). PhANNs outputs the per class sum of the ten ANNs scores (the maximum achievable PhANNs score is 10, as there are ten ANNs). The input sequence is classified as the class with the highest PhANNs score.

To give a clearer indication of the quality of this prediction we added a "confidence" score to each prediction. The "confidence" score shows what fraction of sequences in the test set that were classified as the same class as the input sequence, and with the same PhANNs score or higher, were correctly classified (True positives). The confidence scores differ depending on the protein class. For example, a sequence classified as "major capsid" with a PhANNs score of 7 has 97% confidence, while a "Tail fiber" with a PhANNs score of 7 has only 82.4% confidence. The per class relationship between the PhANNs score and the confidence is explored in **Fig 5**.

## Web server

We developed an easy-to-use web server for users to upload and classify their own sequences. Although ANNs need substantial computational resources for training the model (between 54,861 and 127,756,413 parameters need to be tuned, depending on the model), the trained model can make fast *de novo* predictions. Our web server (https://edwards.sdsu.edu/phanns) can predict the structural class of an arbitrary protein sequence in seconds and assign all the

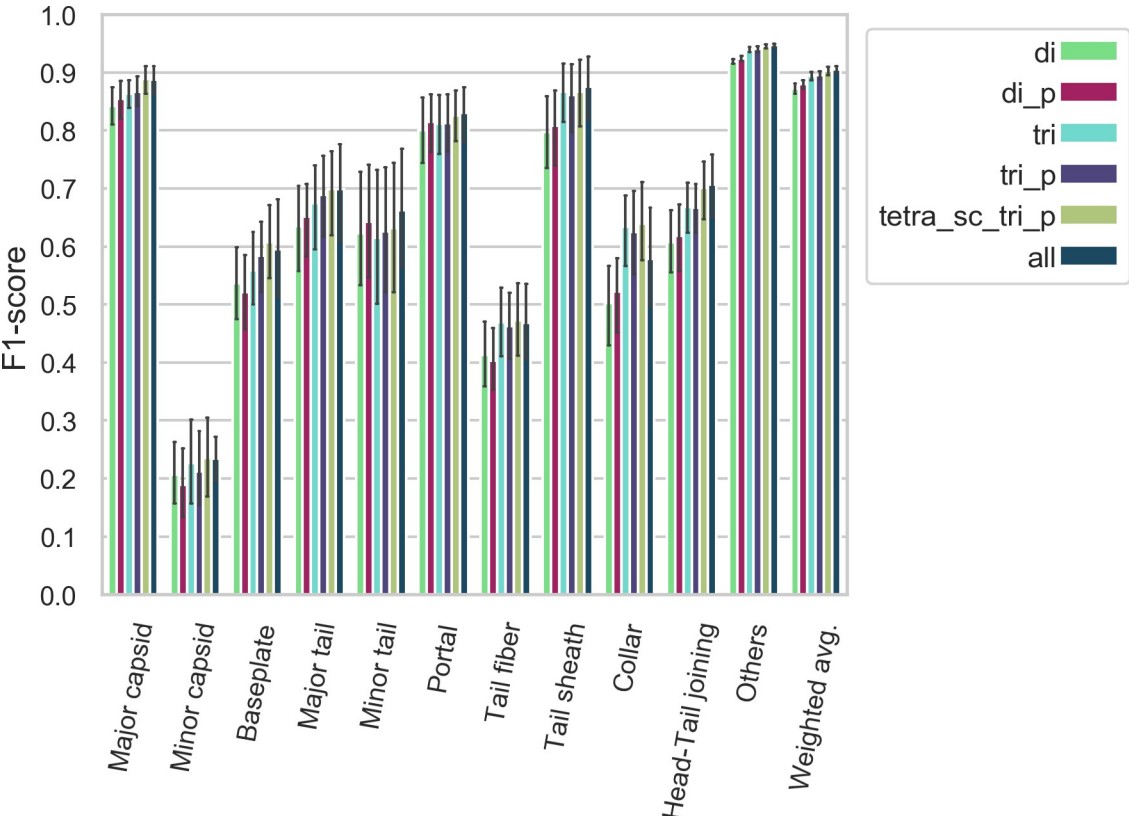

**Fig 3. Class-specific F$_1$ score—F$_1$ scores (harmonic mean of precision and recall) for each polypeptide model/class combination.** Some classes, such as minor capsid, tail fiber, or minor tail, are harder to classify correctly irrespective of the model used. Error bars represent the 95% confidence intervals.

ORFs in a phage genome to one of the 10 classes in minutes. The application can also be downloaded and run locally for large numbers of queries or if privacy is a concern.

## Results and discussion

We evaluated the performance of 120 ANNs (10 per model type) on their respective validation set. For each ANN, we computed the precision, recall, and F$_1$-score of the 11 classes. A "weighted average" precision, recall and F$_1$-score, where the score for each class is weighted by the number of proteins in that class (larger classes contribute more to the score) was computed. The accuracy (fraction of observation correctly classified) is equivalent to the weighted average recall. The three weighted average scores are represented as a 12th class. This gives us ten observations for each combination of model type and class, which allows us to construct the confidence intervals depicted in **Figs 2, 3 and 4.**

(**Figs 2 and S1**) shows that all the models follow the same trend as to which classes they predict with higher or lower accuracy. Some classes of proteins, for example major capsids, collars, and head-tail joining proteins, are predicted with high accuracy. On the other hand, the minor capsid and tail fiber protein classes seem to be intrinsically hard to predict with high accuracy irrespective of the model type used (**Figs 3 and S2**). One reason for this is the limited size of the training set: the minor capsid protein set is the smallest class, with only 581 proteins available for inclusion in our database. Even if the classes were weighted according to their size during training, it appears we do not have enough training examples to identify this class with

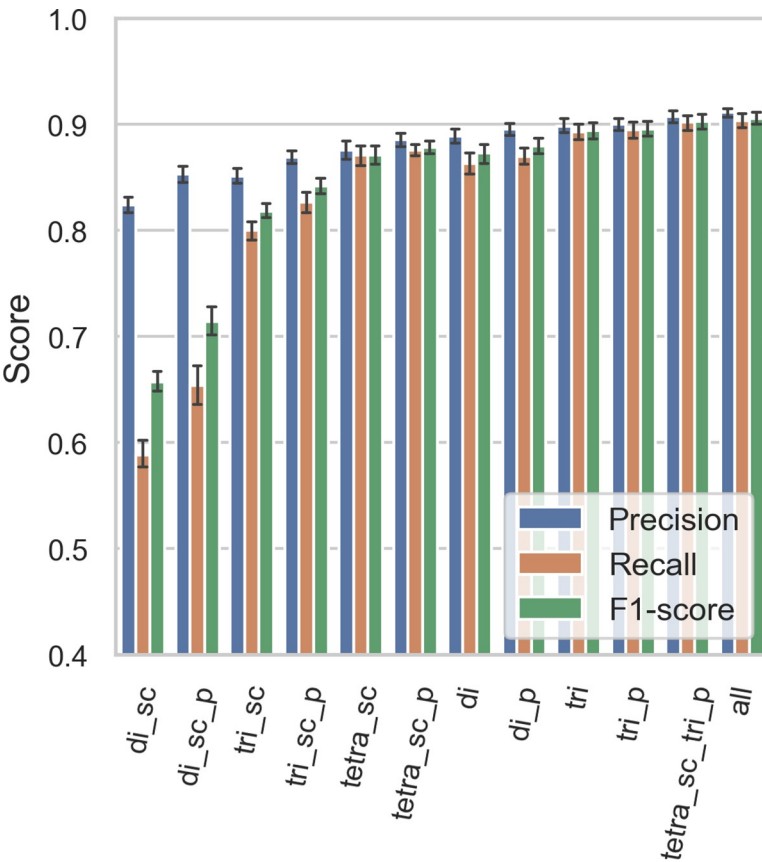

**Fig 4. Model-specific validation weighted average scores—Precision, recall, and F$_1$ scores for all models.** Precision is higher in all models as the "others" class is the largest and easiest to classify correctly. Error bars represent the 95% confidence intervals.

high accuracy. Furthermore, "minor capsid" is often misclassified as "portal" (**Fig 6**). This probably reflects an annotation bias, as we found about 800 proteins annotated as "portal (minor capsid)" in the raw sequences. When the ~800 proteins are analyzed with PhANNs, over 90% are predicted to be portal proteins. Although these were removed during manual curation of the training data sets, some (small) fraction of minor capsid proteins in our database may have been annotated as "minor capsid" by homology to one of those 800 sequences.

The predictive accuracy for a specific class of proteins is likely to be affected by the bias in the training datasets. The bias could be biological and/or due to a sampling bias. An example of the former is the tail fiber class: the tail fiber is one of the determinants of the host range of the virus, and is under strong evolutionary selective pressure [24–29]. On the other hand, sampling bias may be introduced due to oversampling of certain types of phages, such as the thousands of mycobacterial phages isolated as part of the SEA-PHAGES project [30], many of which are highly related to each other.

Average validation F$_1$-scores range from 0.653 for the "di_sc" model to 0.841 for the "tetra_sc_tri" model (**Fig 4**). Although the average validation F$_1$-score for the top three models "tri_p" (0.832), "tetra_sc_tri_p" (0.841), and "all" (0.827) are not significantly different from each other, we decided to use "tetra_sc_tri_p" for the web server and all subsequent analyses because, while it uses ~7% fewer features than "all" (10,409 vs 11,201), we expect that the tetra side chain features may be better than the tripeptide features at generalizing predictions and accessing greater sequence diversity.

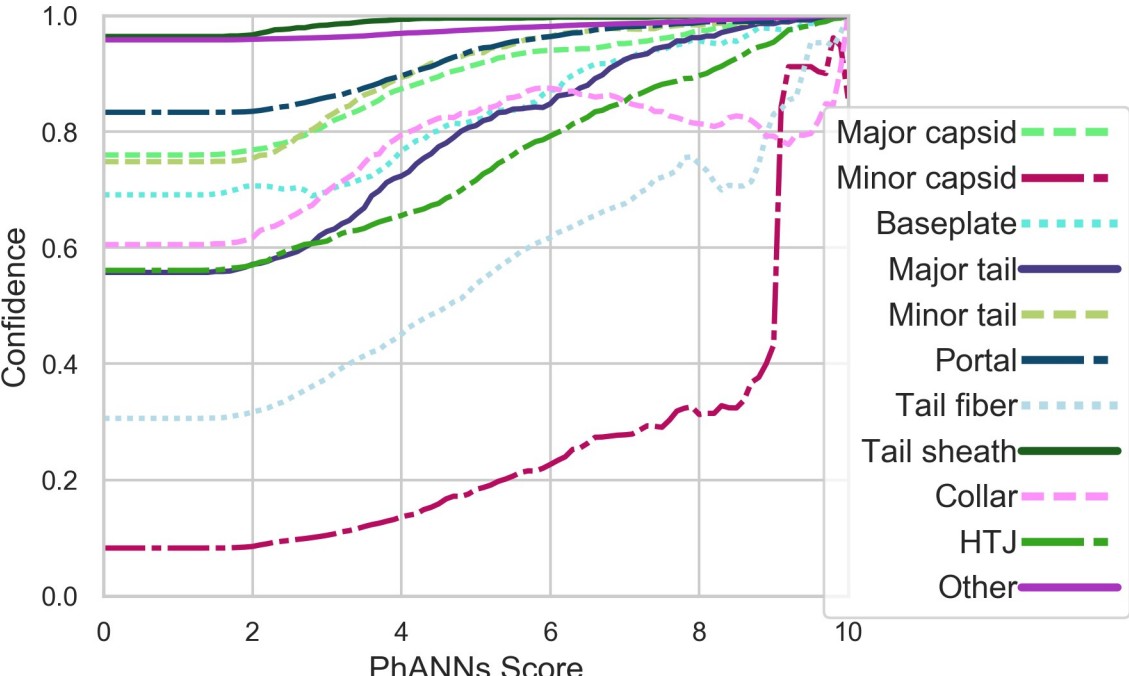

**Fig 5. Per class relationship between PhANNs score and confidence—The confidence corresponding to a particular class PhANNs score represents the fraction of true positives (correctly classified) sequences in the test set that were classified as that class, with a given PhANNs score or higher.** As it is uncommon for the highest class PhANNs score to be less than 2, the left side of the graph includes all test proteins that were classified as that class, and the confidence corresponds to the per class precision (see Table 4).

Using the "tetra_sc_tri_p" ensemble, we predicted the class of each sequence in the test set (46,801) by averaging the scores of each of the ten ANNs. Results are summarized in **Fig 6 and Table 4.** Doing this we reach a test $F_1$-score of 0.89 and accuracy of 86.2% over the eleven classes.

Higher accuracy can be reached if one is willing to disregard sequences with low PhANNs scores. Using only sequences with a PhANNs score of 5 or higher, the $F_1$-score for the test set is 0.945, accuracy is 94%, with 9,006 of 46,801 (~20%) test sequences being "not classified". If using sequences with a PhANNs score of 8 or higher, the $F_1$-score for the test set is 0.982, accuracy is 98%, but 19,208 of 46,801 (~41%) test sequences would be "not classified" (see **Fig 7**). **Table 4** shows summary statistics for the complete test set, while **Table 5** shows the same statistics for the test subset of sequences with PhANNs 8 or greater. The stringency with which users interpret the PhANNs score may vary depending on their specific need. Therefore we recommend that the actual PhANNs score (or the confidence score) be reported in addition to the predicted function class.

Because "minor capsid" is the worst performing class in our test set, we trained an analogous ANN ensemble without that class to explore if accuracy of the remaining classes is improved. Multiple metrics can be used to assess which model is better. The per class ROC curves of both models [**Fig 8A (with minor capsid class)** and 8-B **(without minor capsid class)**] and areas under the curves are similar. Removing the minor capsid class from the models doesn't significantly alter the relationship between the PhANNs score and the confidence score (**Fig 8C and 8D**). The confusion matrices of both models (**Fig 8E and 8F**) show that predictions for portal proteins improve, as 3% of them are misclassified as minor capsid. For all other classes, the two models are similar with respect to which classes are most commonly

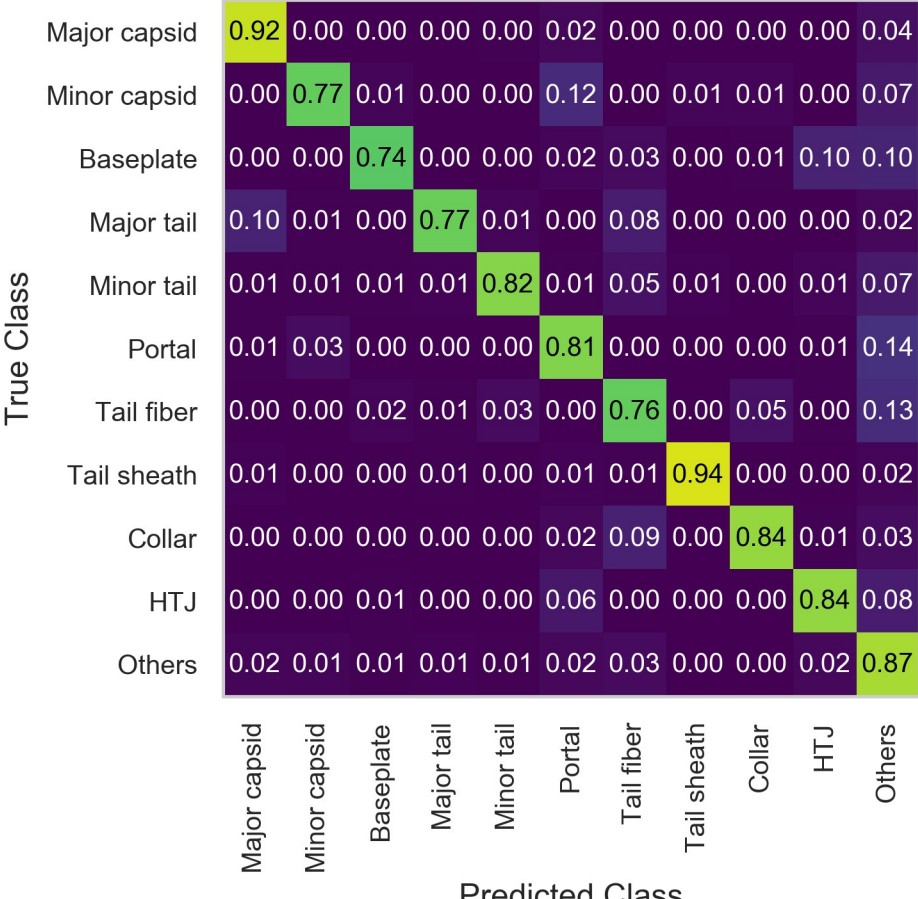

**Fig 6. Confusion matrix using the "tetra_sc_tri_p" model—Each row shows the proportional classification of test sequences from a particular class.** A perfect classifier would have 1 on the diagonal and 0 elsewhere. In general, a protein that is misclassified is predicted as "others".

**Table 4. Results of per class classification for the test set.** Support indicates the number of test sequences in each specific class. accuracy (fraction of observation correctly classified) is equivalent to the weighted average recall (weighted by the support of each class). The macro average is unweighted (all classes contribute the same).

|  | precision | recall | f1-score | support |
|---|---|---|---|---|
| Major capsid | 0.80 | 0.91 | 0.85 | 2,456 |
| Minor capsid | 0.07 | 0.78 | 0.13 | 81 |
| Baseplate | 0.69 | 0.75 | 0.72 | 851 |
| Major tail | 0.55 | 0.79 | 0.65 | 502 |
| Minor tail | 0.66 | 0.82 | 0.73 | 1,072 |
| Portal | 0.81 | 0.81 | 0.81 | 5,261 |
| Tail fiber | 0.35 | 0.74 | 0.47 | 648 |
| Tail sheath | 0.97 | 0.93 | 0.95 | 2,031 |
| Collar | 0.51 | 0.86 | 0.64 | 300 |
| Head-Tail joining | 0.56 | 0.84 | 0.67 | 1,277 |
| Others | 0.96 | 0.86 | 0.91 | 32,322 |
| macro avg | 0.63 | 0.83 | 0.68 | 46,801 |
| weighted avg | 0.89 | 0.86 (accuracy) | 0.87 | 46,801 |

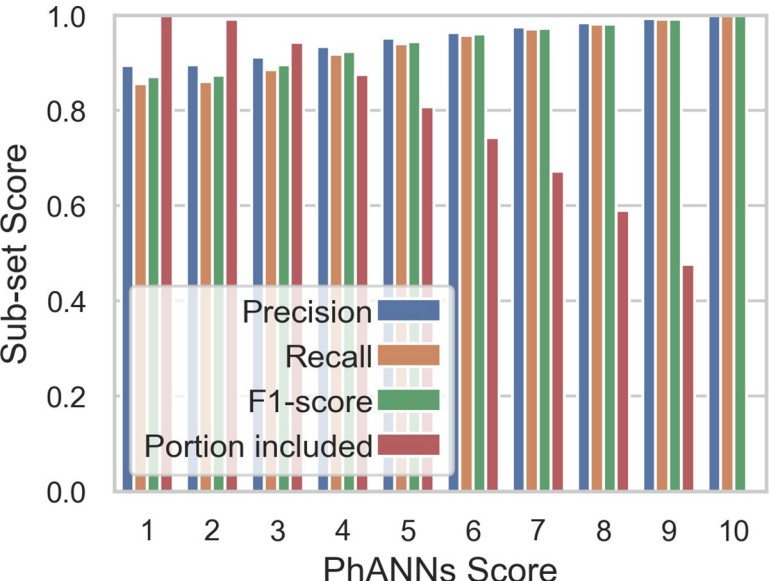

**Fig 7. Effect of disregarding low scoring test proteins—Progression of the weighted average precision, recall and F$_1$-score of the test set after excluding low scoring proteins.** The portion of included proteins is the fraction that can be classified if you only trust that score or higher. Very few test proteins have PhANNs score of 10 and not all classes are represented.

confused. A comparison of per class precision, recall and F$_1$-score can be found in **Table 6.** When the minor capsid class is excluded, metrics are just as likely to improve as to worsen, and the accuracy gain is only 1%; greater accuracy gains can be achieved by disregarding sequences with low PhANNs scores as "not classified," as described above. Therefore, we decided not to exclude the minor capsid class from our model; the performance in this class is likely to improve in the future, as more sequences become available and, hopefully, are experimentally validated.

**Table 5. Results of per class classification for proteins in the test set with a PhANNs score of 8 or higher.** Support indicates the number of test sequences in each specific class. accuracy (fraction of observation correctly classified) is equivalent to the weighted average recall (weighted by the support of each class). The macro average is unweighted (all classes contribute the same).

|  | precision | recall | F$_1$-score | support |
|---|---|---|---|---|
| Major capsid | 0.99 | 0.99 | 0.99 | 1,563 |
| Minor capsid | 0.28 | 0.96 | 0.43 | 45 |
| Baseplate | 0.97 | 0.83 | 0.89 | 151 |
| Major tail | 0.95 | 0.97 | 0.96 | 307 |
| Minor tail | 0.95 | 0.99 | 0.97 | 625 |
| Portal | 0.99 | 0.94 | 0.97 | 3,810 |
| Tail fiber | 0.89 | 0.94 | 0.91 | 360 |
| Tail sheath | 1.00 | 1.00 | 1.00 | 1,495 |
| Collar | 0.82 | 1.00 | 0.90 | 98 |
| Head-Tail joining | 0.91 | 1.00 | 0.95 | 916 |
| Others | 0.99 | 0.99 | 0.99 | 18,223 |
| macro avg | 0.89 | 0.96 | 0.91 | 27,593 |
| weighted avg | 0.98 | 0.98 (accuracy) | 0.98 | 27,593 |

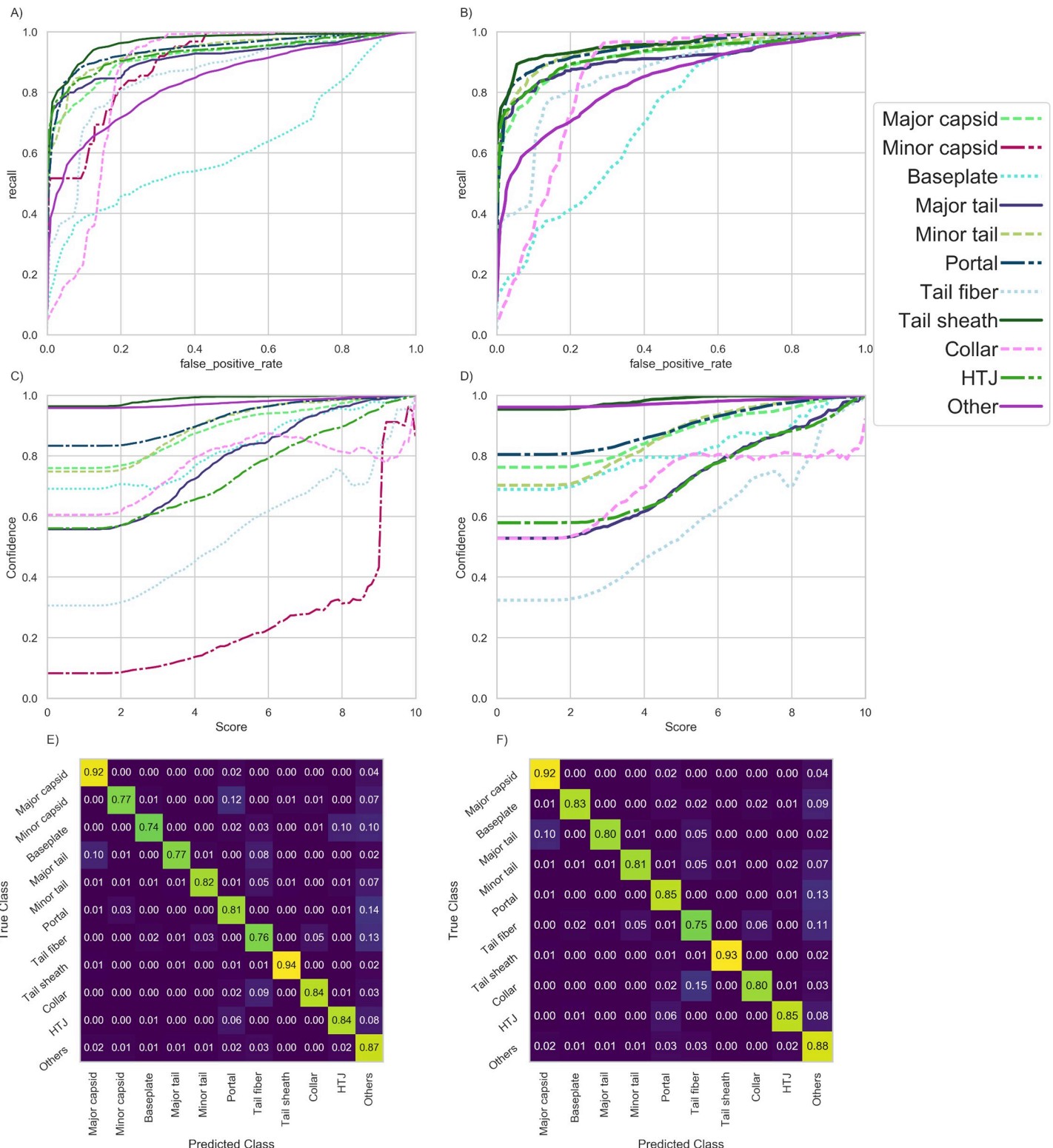

**Fig 8. Comparison of "tetra_sc_tri_p" model trained with and without the Minor capsid class—As minor capsid is the worst performing class in our test set, we trained an analogous ANN ensemble with it removed.** Panels A and B show the ROC curves for the models with and without minor capsid respectively. Panels C and D show the relationship between PhANNs score and Confidence for the models with and without minor capsid respectively. Panels E and F show the confusion matrix for the models with and without minor capsid respectively.

**Table 6. The effect on the models's scores from excluding the minor capsid class (mc)—Most scores are affected only slightly and are as likely to improve as to worsen.**

|  | precision | precision (mc) | recall | recall (mc) | F$_1$-score | F$_1$-score (mc) | support | ROC area | ROC area (mc) |
|---|---|---|---|---|---|---|---|---|---|
| **Major capsid** | 0.76 | 0.76 | 0.92 | 0.92 | 0.83 | 0.83 | 2456 | 0.917 | 0.918 |
| **Minor capsid** | 0.08 | - | 0.77 | - | 0.15 | - | 81 (0) | 0.899 | - |
| **Baseplate** | 0.69 | 0.69 | 0.74 | 0.83 | 0.72 | 0.75 | 851 | 0.621 | 0.72 |
| **Major tail** | 0.56 | 0.53 | 0.77 | 0.80 | 0.65 | 0.64 | 502 | 0.918 | 0.91 |
| **Minor tail** | 0.75 | 0.70 | 0.82 | 0.81 | 0.78 | 0.75 | 1070 | 0.939 | 0.94 |
| **Portal** | 0.83 | 0.80 | 0.81 | 0.85 | 0.82 | 0.82 | 5261 | 0.943 | 0.945 |
| **Tail fiber** | 0.31 | 0.32 | 0.76 | 0.75 | 0.44 | 0.45 | 648 | 0.861 | 0.86 |
| **Tail sheath** | 0.96 | 0.95 | 0.94 | 0.93 | 0.95 | 0.94 | 2031 | 0.986 | 0.957 |
| **Collar** | 0.61 | 0.53 | 0.84 | 0.80 | 0.70 | 0.63 | 300 | 0.865 | 0.85 |
| **HTJ** | 0.56 | 0.58 | 0.84 | 0.85 | 0.67 | 0.69 | 1277 | 0.933 | 0.923 |
| **Others** | 0.96 | 0.96 | 0.87 | 0.88 | 0.91 | 0.92 | 33402 | 0.838 | 0.838 |
| **macro avg** | 0.64 | 0.68 | 0.83 | 0.84 | 0.69 | 0.74 | 47879 (47798) |  |  |
| **weighted avg** | 0.90 | 0.90 | 0.86 | 0.87 | 0.88 | 0.88 | 47879 (47798) |  |  |

We compared the performance of PhANNs with that of VIRALpro by predicting the function class of each other's test set. Doing this requires us to map our 11 classes onto VIRALpro's 4 (capsid versus not-capsid, tail versus not tail). We decided not to use the PhANNs "collar" or "baseplate" test set as VIRALpro has a hard time classifying them (presumably because it was not trained on those classes). Hence we discarded any of the VIRALpro test proteins that PhANNs predicted as "collar" or "baseplate". "Capsid" in VIRALpro means either "major capsid" or "minor capsid" in PhANNs. "Tail" in VIRALpro means "Major tail", "Minor tail", "Tail fiber" or "Tail sheath" in PhANNs. This transformation makes possible the comparison of the two tools. Results are summarized in **Table 7**. The two tools have similar accuracy, with VIRALpro slightly better at predicting capsid proteins and PhANNs slightly better at predicting tail proteins. It is important to mention that the VIRALpro predictions took several days on a 200+ CPU cluster (it would take a few years on a laptop). A similarly sized test takes less than an hour using the PhANNs server.

The utility of the PhANNs tool is to permit more extensive function predictions of metagenome sequences from phages used for phage therapy (A. Cobian, N. Jacobson, M. Rojas, H. Hamza, R. Rowe, D. Conrad, and A. Segall, et al., work in progress) and to better describe the coding potential of the virome in patients suffering from diseases such as inflammatory bowel disease versus household controls (A. Segall, R. Edwards, A. Cantu, S. Handley, and D. Wang, work in progress). In some cases, phage-associated sequences from isolated viromes have no or very weak functional predictions when using BLAST, RPS-BLAST, or related bioinformatic tools (work in progress). In parallel, we are experimentally validating some of the predicted functions using electron microscopy and X-ray crystallography (S.H. Hung, V. Seguritan, et al., ms. in preparation).

**Table 7. Comparison of PhANNs with VIRALpro. Results from using VIRALpro test set in PhANNs and PhANNs test set in VIRALpro.**

|  | PhANNs test set in TAILpro | TAILpro test set in PhANNs | PhANNs test set in CAPSIDpro | CAPSIDpro test set in PhANNs |
|---|---|---|---|---|
| test set size | 10,805 | 672 | 15,107 | 787 |
| precision | 0.28 | 0.77 | 0.14 | 0.82 |
| recall | 0.79 | 0.68 | 0.86 | 0.32 |
| accuracy | 0.80 | 0.82 | 0.70 | 0.67 |
| F1-score | 0.42 | 0.72 | 0.25 | 0.46 |

The performance of any machine learning system is limited by the availability and cost of training examples [14]. Invariably, top performing image and audio classification systems must augment their training data with synthetic examples created by applying semantically orthogonal transformations to the training set (i.e., slightly rotating or distorting an image, or adding background noise to audio) [31,32]. In bioinformatics, the current practice of de-replication moves us in exactly the opposite direction—perfectly good samples cannot be used if their overlap with other samples is too high, leaving only one version of the biostring to use for training, thereby ignoring sequence variations. This despite the fact that biological examples such as protein sequence data are replete with variations from a consensus sequence or motif. Our approach overcomes this failing by using *all* non-redundant data. By splitting the dataset into the training, validation, and test sets after first de-replicating at 40%, we remove even slightly redundant samples and make sure that none of the performance is due to data memorization rather than generalization. Augmenting the training set by expanding the clusters to include all non-redundant samples is the novel idea we have introduced in the present paper as a way of increasing our training set size and hence our accuracy.

## Conclusion

ANNs are a powerful tool to classify phage structural proteins when homology-based alignments do not provide useful functional predictions, such as "hypothetical" or "unknown function". This approach will become more accurate as more and better characterized phage structural protein sequences, especially more divergent ones, are experimentally validated and become available for inclusion in our training sets. This method can also be applied to predicting the function of unknown proteins of prophage origin in bacterial genomes. In the future, we plan to expand this approach to more protein classes and to viruses of eukaryotes and archaea.

## Supporting information

**S1 Table. Side chain groupings.**
(XLS)

**S1 Fig. Model-specific $F_1$ score—$F_1$ scores (harmonic mean of precision and recall) for each side chain model/class combination.** All models follow similar trends as to which classes are more or less difficult to classify correctly. Error bars represent the 95% confidence intervals.
(PNG)

**S2 Fig. Class-specific $F_1$ score—$F_1$ scores (harmonic mean of precision and recall) for each side chain model/class combination.** Some classes, such as minor capsid, tail fiber, or minor tail, are harder to classify correctly irrespective of the model used. Error bars represent the 95% confidence intervals.
(PNG)

**S3 Fig. Comparison of the validation weighted average $F_1$-score of three models on the same feature sets—We compared our ANN ensemble trained on 1D-10D sets against a logistic regression trained on the 1D-10D sets and an ANN ensemble trained on the 1d-10d sets (40% dereplication, without cluster expansion—see Methods).** The ANN ensembles perform significantly better than the logistic regression. Error bars represent 0.95 confidence intervals.
(PNG)

**S4 Fig. Per class comparison of the validation $F_1$-score of three models on the "tetra-s_sc_tri_p feature" set—In the structural classes, the 1D-10D ANN ensemble performs slightly better than the logistic regression and significantly better than the 1d-10d ANN ensemble.** In the "others" class (by far the largest), 1D-10D ANN ensemble performs as well as 1d-10d ANN and better than logistic regression. Error bars represent 0.95 confidence intervals.
(PNG)

## Acknowledgments

AMS would like to acknowledge Drs. Sherwood Casjens (University of Utah) and Ian Molineux (University of Texas Austin) for helpful conversations on phage biology.

## Author Contributions

**Conceptualization:** Vito Adrian Cantu, Peter Salamon, Victor Seguritan, Jackson Redfield, Robert A. Edwards, Anca M. Segall.

**Data curation:** Vito Adrian Cantu, Robert A. Edwards, Anca M. Segall.

**Formal analysis:** Vito Adrian Cantu, Peter Salamon, Robert A. Edwards.

**Funding acquisition:** Robert A. Edwards, Anca M. Segall.

**Investigation:** Vito Adrian Cantu, Anca M. Segall.

**Methodology:** Vito Adrian Cantu, Peter Salamon, David Salamon, Robert A. Edwards.

**Project administration:** Anca M. Segall.

**Resources:** Robert A. Edwards, Anca M. Segall.

**Software:** Vito Adrian Cantu, Robert A. Edwards.

**Supervision:** Peter Salamon, Anca M. Segall.

**Validation:** Vito Adrian Cantu, Peter Salamon.

**Visualization:** Vito Adrian Cantu.

**Writing – original draft:** Vito Adrian Cantu, Peter Salamon, Anca M. Segall.

**Writing – review & editing:** Vito Adrian Cantu, Peter Salamon, David Salamon, Robert A. Edwards, Anca M. Segall.

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
