## [Decision Letter · Decision Letter 0]

16 May 2020

Dear Mr. Cantu Alessio Robles,

Thank you very much for submitting your manuscript "PhANNs, a fast and accurate tool and web server to classify phage structural proteins" for consideration at PLOS Computational Biology.

As with all papers reviewed by the journal, your manuscript was reviewed by members of the editorial board and by several independent reviewers. In light of the reviews (below this email), we would like to invite the resubmission of a significantly-revised version that takes into account the reviewers' comments.

We cannot make any decision about publication until we have seen the revised manuscript and your response to the reviewers' comments. Your revised manuscript is also likely to be sent to reviewers for further evaluation.

Sincerely,

Mihaela Pertea

Software Editor

PLOS Computational Biology

Mihaela Pertea

Software Editor

PLOS Computational Biology

Reviewer's Responses to Questions

**Comments to the Authors:**

Reviewer #1: Cantu et al. present the tool PhANNs for the prediction of phage genes. This tool is greatly needed as the number of phage genomes being detected in metagenomic studies continues to increase and the majority of the genes are unable to be annotated. Having now tested the tool, my comments re: the manuscript can be divided into (1) the tool/code repo and (2) the manuscript. Overall, the manuscript is well written and clear.

Tool:

1. Question: Both the Webserver and GitHub repo rely on this manually-curated database. As is a concern frequently with new tools, how do the authors plan to keep this database up to date?

2. Question: It looks like there is a way for the user/community to update the model file and/or create their own. For building the database and model, I'm assuming it's in the model_training folder, but there is no ReadMe describing the files in there.

3. The web service results list a table. It's great to be able to download it, but there is no details on how to interpret this table. What numbers are being reported by the web? Also, why are some cells highlighted on the web results? Does it automatically pick the highest score or is there some way that it distinguishes a good prediction? This information should be provided on the webpage, either with the results or the submission page. The web service is an attractive tool for less computational users. It however lacks documentation.

4. As a test to see how well the tool worked with non-phage sequences, I fed it some bacterial protein sequences and was surprised to find predictions other than "Other". Some had scores of 10.00, of course without knowing what the number being reported is and what the range is, is that good? I'm imagining a scenario where a user has a contig they think is viral but are uncertain.

Manuscript:

1. Details re: the prediction models are not thoroughly discussed. As this is the computational contribution of this work, additional details and testing would strengthen the paper. Related, I would consider moving Table S2 into the main text. There is no other mention of the codes used for the model names and what the individual models consider in the main text. This makes interpretation of Figure 2 challenging. Alternatively, the Figure 2 legend should include details re: what each of the model names listed on the x axis mean.

2. While the authors acknowledge that there are other tools for phage function prediction, the manuscript lacks a comparison to these tools. For instance, Galiez et al. has a better accuracy than PhANNs for capsid and tail predictions (as indicated in Table 1). And while yes, PhANNs is considering more than just these two types of phage proteins, a comparison of how PhANNs performs relative to these other solutions would convince the reader that they should use PhANNs. Or perhaps they should use another tool for specifically capsid genes in addition to PhANNs. How does PhANNs predictions compare to these other tools? Is it classifying the same proteins?

Reviewer #2: The manuscript by Cantu et al. presents a method for classifying proteins into 11 categories: 10 phage structural types and one catch-all ‘other’ group. The method applies neural network training, and used a large number of protein sequences to generate 12 models. Their source code is freely available under the MIT license for anonymous download in Github, and there is a webserver that applies one of those models (tetra_sc_tri_p) to user data. This work extends upon previous work by the same group and is meant to fill a need in the field for quick classification methods of unstudied phage proteins with low likelihood to be anything besides phage structural proteins. There are multiple concerns, however, around the generation of the training set and validation of the resulting models.

GENERAL COMMENTS

1. The training data sets for the neural network models were collected from the NCBI database using the ncbi_get_structural.py script, which retrieves sequences from the NCBI protein database based solely on how they are named by the sequence depositors. This is problematic, as a) many phage protein annotations in the database are incorrect, and b) even if they are “correct” there is no enforceable naming convention between annotators. For example, the term “minor tail protein” could refer to baseplate components, tail needles, tail fibers, tail tape measure proteins or head-tail joining proteins. Likewise, some groups may use alternate names for the major capsid protein, such as “coat” or “MCP”, and some names may simply be misspelled. It should also be noted that the ncbi_get_structural.py script contains a typo at line 145 (“head-tail joinning”) that would have interfered with data retrieval.

2. Manual curation is mentioned in the manuscript (lines 25, 83, 93, Table 2), but no curation criteria for class inclusion or exclusion are provided. Did the curation involve any validation that the proteins in a given class (e.g., tail fiber) actually had their named functions? Some categories, like baseplate, decreased significantly from manual curation (Table 2) but we do not know why.

3. The biological or structural definitions of each of the ten classes are not defined in the manuscript, so it is difficult to tell what structural components belong to each class. While terms such as “major capsid” and “portal” are generally well-agreed upon (and will often contain those terms in their name), other categories are more ambiguous. Aside from the ambiguity in what constitutes a “minor tail protein” described above, it is not clear what qualifies as “major tail” (could be tail tube, tail sheath or both), a “tail shaft”, or what the structural distinction is between “collar” and “head-tail joining” proteins. The authors are free to define the classes as they wish, but detailed definitions are required, especially if the intent is for these annotations to be applied to novel phage genomes.

4. The manuscript describes validation of the neural network against sets of database proteins, but there is no validation of the tool against sets of known, experimentally verified proteins, or against clusters of structural proteins that are supported by external bioinformatic evidence (e.g., conserved domains, COGs, high-quality UniProt/SwissProt annotations). The authors describe this as a tool for the annotation of novel phage genomes, so there must be some evidence that the tool “works” on predicting known proteins. This validation should also include a description of the scoring criteria produced in the tool output, which appears to score each protein for each category on a scale of 0 to 10. It is not clear how to interpret these outputs or what constitutes a “good” score for the tool.

SPECIFIC COMMENTS

Line 43 – ‘by lysing specific components of microbiomes’ is unclear. Is it meant that specific taxa are targeted? Phage can also interact with bacteria in ways other than lysing them.

Lines 60-62 – the argument that phage therapy has increased demand for phage annotation is supported, but the implication here seems to be that annotation of structural proteins will provide higher confidence in using phages for therapy. Is there a specific connection (positive or negative) to structural proteins and phages used for therapy? What does “provisional” mean in this context? Is this just another word for “inaccurate”?

Figs. 2, 3 and 4 are missing y-axis labels. The bars in Figs. 2 and 3 are very narrow and can only be viewed when zoomed in to >200%.

**Have all data underlying the figures and results presented in the manuscript been provided?**

Reviewer #1: Yes

Reviewer #2: Yes

PLOS authors have the option to publish the peer review history of their article (what does this mean?). If published, this will include your full peer review and any attached files.

Reviewer #1: No

Reviewer #2: No
---

## [Decision Letter · Decision Letter 1]

26 Sep 2020

Dear Mr. Cantu Alessio Robles,

We are pleased to inform you that your manuscript 'PhANNs, a fast and accurate tool and web server to classify phage structural proteins' has been provisionally accepted for publication in PLOS Computational Biology.

Best regards,

Mihaela Pertea

Software Editor

PLOS Computational Biology

Mihaela Pertea

Software Editor

PLOS Computational Biology

Reviewer's Responses to Questions

**Comments to the Authors:**

Reviewer #1: The authors have addressed all of the reviewer comments in their revised manuscript. The gitHub repo and tool are both easy to understand and work well. The additional figures and tables provide additional support for the utility and power of this tool.

Reviewer #2: I think the authors for their attention to the reviewer comments.

**Have all data underlying the figures and results presented in the manuscript been provided?**

Reviewer #1: Yes

Reviewer #2: Yes

PLOS authors have the option to publish the peer review history of their article (what does this mean?). If published, this will include your full peer review and any attached files.

Reviewer #1: **Yes: **Catherine Putonti

Reviewer #2: No